# The impact of occupational structures on ethnic and gendered employment gaps: An event history analysis using social security register data

Stefan Vogtenhuber[1,2]*, Nadia Steiber[1,2], Monika Mühlböck[2], Bernhard Kittel[2]

**1** Department of Sociology, Institute for Advanced Studies, Vienna, Austria, **2** Department of Economic Sociology, University of Vienna, Vienna, Austria

☉ These authors contributed equally to this work.
* vogten@ihs.ac.at

## Abstract

Ethnic and gendered employment gaps are mainly explained by individual characteristics, while less attention is paid to occupational structures. Drawing on administrative data, this article analyses the impact of occupational characteristics on top of individual attributes in the urban labour market of Vienna. Both set of variables can explain observed employment gaps to a large extent, but persistent gaps remain, in particular among females. The article's main finding is that the occupational structure appears to have gendered effects. While men tend to benefit from ethnic segregation, women face difficulties when looking for jobs with high shares of immigrant workers. Looking for jobs in occupations that recruit from relatively few educational backgrounds (credentials) is beneficial for both sexes at the outset unemployment, but among females this competitive advantage diminishes over time. The article concludes by discussing potential strategies to avoid the traps of occupational segregation.

## Introduction

Employment gaps between newcomers to Western labour markets and their resident populations are well-established. Compared to natives, immigrants tend to be less active on the labour market, face higher unemployment risks, and are disproportionally employed in low-wage and low-prestige economic sectors [e.g. 1]. There are, however, systematic differences in the extent to which newcomers can structurally integrate across origin countries and gender. Persistent employment gaps have been observed among immigrants from Turkey in Austria, Belgium, Germany and the Netherlands, from Tunisia in Belgium, France and the Netherlands, as well as immigrants from Pakistan in Britain [2]. There is a pronounced gender dimension to these gaps, as they are generally much higher among females than males [3].

The explanation of employment gaps has focused on individual human capital characteristics [for a review see 2], while less attention has been paid to structural factors embedded in labour market institutions. We contribute to the literature on ethnic and gendered

accessing the pseudonymised microdata used in this study can be filed under https://arbeitsmarktdatenbank.at/ following registration. Data requests may also be sent to Mr. Andreas Buzek (Andreas.Buzek@bmafj.gv.at) of the Federal Ministry of Labour, Family and Youth.

**Funding:** All authors received funding from the Austrian Federal Ministry of Labour, Social Affairs and Consumer Protection as part of the JuSAW project. SV and NS were supported by funds of the Oesterreichische Nationalbank [Anniversary Fund, project number: 17177]. The funders had no role in study design, data collection and analysis, decision to publish, or preparation of the manuscript.

**Competing interests:** The authors have declared that no competing interests exist.

employment gaps by analysing the influence of structural opportunities on top of individual characteristics. Drawing on comprehensive administrative data we analyse the time it takes young people who have become unemployed to find new employment in the urban labour market of the Austrian capital Vienna. Our data covers all employment and unemployment spells of all Viennese residents aged 18 to 28 years before and after they have newly registered as unemployed with the Public Employment Service (PES) within the time frame April to September 2014 (time 0). The data thus allows us to monitor their employment trajectories while at the same time accounting for a host of individual characteristics relevant to job search and labour market placement, including education, prior work experience, immigration status and nationality of origin. As a structural factor we focus on occupations in which the young unemployed are looking for new jobs because in the context of the occupation-specific Austrian labour market and employment regime they shape—contingent on individual characteristics—structural opportunities that facilitate or hinder employment prospects. In particular, the impact of three occupational characteristics are analysed. First, occupational closure is the extent to which occupations recruit from varying educational credentials. Second, ethnic segregation is the extent to which jobs are occupied by natives or immigrants and third, gender segregation is the extent to which occupations are dominated by a particular gender. Like many Western European cities, the Viennese population is highly dynamic and its labour force is growing mainly due to immigration and increasing female participation.

The article is organised as follows. The next section discusses current strands in the literature before laying out our own framework that integrates ethnic, gender and structural occupational determinants in explaining employment gaps. Before introducing data, variables and methods we show how past immigration changed labour supply in Vienna. Then we present our findings, which are then discussed in the concluding section.

## Theoretical framework

### Individual level explanations

Differences in human capital endowments are among the most prominent individual-level explanations of employment gaps. Education and on-the-job training enhance productive capacities and should therefore increase employment probabilities [4]. Job seekers who lack formal qualifications, job-specific skills and experience demanded by the labour market have limited employment prospects. Other explanations refer to social and cultural characteristics. A brief discussion of theoretical accounts with regard to ethnicity (i.e. nationality) and gender follows.

Missing language skills are an important barrier to structural integration because they prevent newcomers to fully utilise their human capital. The limited portability of human capital may also arise from country differences in the demand for skills. Education, skills and experience that have been accumulated in origin countries may very well differ from the ones in demand by the host country's labour market. Although international migrants are generally positively selected in terms of human capital [5], there is a large gap in educational attainment levels between immigrants to EU countries who have completed their education outside the EU and the native-born population. 54% of adult immigrants from third countries have completed only lower-secondary education or less, compared to 27% of the native born [6]. A recent survey found similar high shares of low education levels among recent refugees in Austria [7]. The authors observe a large gap in previous work experience between men (90%) and women (42%). There is evidence that the share of low-educated immigrants is larger among women than among men in European countries [8,9]. Adsera and Chiswick [8] find that women benefit more from education in terms of labour market outcomes.

In addition to human capital characteristics, socio-cultural aspects are important in many work contexts, where it is relevant that "employees have some familiarity and empathy with the mores and lifestyles of the social groups from which customers or clients are chiefly drawn" [10]. Cultural knowledge may also help to get along with ones' co-workers and supervisors. Studies suggest that familiarity with labour market customs, outgroup contacts, and gender role attitudes influence employment prospects [11]. The gender division of labour within families resulting from more traditional gender role attitudes are associated with a lower economic activity among women [3,12]. According to the World Values Survey, a majority of both male and female respondents from Middle Eastern countries agree with the statement "When jobs are scarce, men should have more rights to a job than women." and only a minority disagrees, while in Western European countries it is the other way around [7]. The gender gap among immigrants from certain origin countries in Western labour markets mirrors the situation in the origin country, especially in the beginning [13]. A low female labour supply in the origin country translates into a low supply in the host country [14].

We thus expect that Employment probabilities among young unemployed job-seekers in Vienna differ across ethnic groups, with natives on top and, depending on the country of origin, female immigrants at the bottom. Moreover, we do expect that the gender gap in the time to employment systematically differs across origin countries, with relatively small gender differences among natives and Eastern European countries and relatively large differences among immigrants from Middle Eastern countries.

## Structural occupational determinants

Beyond individual characteristics, occupational segregation explain much of the employment gaps across ethnicity and gender [15–17]. In this section we investigate potential effects of structural characteristics.

Labour market segmentation theory identifies two basic segments that differ regarding job quality, wages, further training and career advancement opportunities [18]. In the primary internal segment jobs tend to score high on these characteristics, and low in the secondary segment. The occupation-specific segment is considered as an additional segment of major importance in countries characterised by occupational labour markets, like Germany and Austria, [19]. In this segment, the market mechanism is limited, as vacancies require *certified* occupational skills, which reduce the number of eligible candidates. This institutional feature can be described as a mechanism of occupational closure [20].

The key closure device is educational credentialing where the skills certified are closely related to occupational demands. Credentials earned from apprenticeship training and vocational schools provide occupation-specific signals to a host of occupations which should be beneficial in the queue of applicants for such jobs [21]. In the regulated or licensed professions, it is even a necessity to hold a particular credential which carry occupational entitlements. It is important to note that in occupational labour markets, credentialing does not only apply to the typical professions like lawyers, medical professionals, teachers, and the like, but also to skilled crafts and trades occupations that are selective on education and skills [22]. The crafts have maintained historical ties to medieval guilds and in addition to the skills there is a strong occupational self-awareness observable in distinct occupational cultures [23]. As the credentials signal skills *and* cultural codes, they shape the prospects of finding employment, limit competition to "eligibles" and establish barriers to job mobility.

Matching processes in real labour markets never are mechanistic, applicants may apply for positions in various occupational fields other than they are specifically trained for. However, those who are specifically trained for a vacant job and possess professional experience, will

enjoy competitive advantage because employers expect them to be readily productive without the need for much additional training. The advantage will be the higher, the larger the number of rival job-seekers and the lower the number of vacancies. We assess the formal accessibility of occupations by the extent to which they recruit workers with similar (high closure) or diverse (low closure) educational credentials. In recent research the linkage strength between education and employment has been measured either using segregation [24,25] or concentration metrics. These measures account for the fact that low-prestigious occupations may have a high closure, as mentioned before. Occupational labour markets require specific skills at all skill levels with the only exception of the unskilled segment, which is a minor segment of the Viennese labour market as only 7.4% of the workers are employed in elementary occupations [ISCO 9, see 26]. Unemployed hairdressers or house painters, for example, are skilled workers who hold an apprenticeship certificate, which is an asset when applying for a job. They are thus enjoying a similar competitive advantage as IT-professionals or accountants when looking for jobs in their occupations, compared to applicants who are not specifically trained. Our measure thus is not about the level or prestige of credentials but about the linkage-strength between occupations and education independent of the education level attained.

As a testable hypothesis, we expect that occupational closure affects employment positively because job competition is limited to those who possess occupation specific skills (Hypothesis 1).

Occupational segregation along ethnicity and gender may also affect employment chances. Segmented-labour-market theory posits, in line with classical assimilation theory, that ethnic segregation in the labour market is a structural barrier to integration because it limits opportunities of immigrants to low-paying jobs in undesirable segments of the labour market which tend to be abandoned by the native-born [15,27,28]. According to the ethnic economy hypothesis immigrants may avoid these disadvantages by utilising social capital through established immigrant-networks [29]. Sanders and Nee [30] find that the ethnic economy argument has merits in the case of self-employment, but that job opportunities among immigrant employees are better explained by assimilation theory. Kogan and Kalter [31] do not find clear evidence for a positive ethnic economy effect in the Austrian labour market as well, while their findings regarding the job prospects of Turkish and ex-Yugoslav workers suggest a queuing mechanism in which they are ranked according to the employer's relative ethnic preference. On top of credentials, job applicants in this view are ranked according to nationality with native Austrians on the top, followed by various immigrant groups. The relative disadvantage immigrant workers face in job competition may then be high in jobs with few immigrants. However, job queues in occupations with a high share of immigrant workers tend to be disproportionally congested because recent demographic trends brought a stark increase in the immigrant workforce resulting in high unemployment among immigrants (see next chapter). In occupations with a shortage of skilled labour and where employers have a hard time to fill vacancies, ethnic recruiting preferences may not be feasible and sustainable. Moreover, the extent to which European employers discriminate between native and foreign-born applicants as established by correspondence experiments does not warrant the conclusion that the majority or even a large share of employers actually adhere to such hiring practices [32]. We thus hypothesize that while some employers may rank applicants by ethnic origin the prospects of landing a job is highly influenced by the mere number of rivalling job seekers. In this sense, job competition will be high in occupations at the bottom of the labour market with large shares of foreign workers and a low skills specificity. Indeed, in occupations with a large share of immigrant workers employers may privilege the occasional native applicant when there is a premium on a minimum of diversity. The same may apply to jobs at the top where most of the workers are natives and minority workers may be welcome in order to increase diversity. Still, as native

workers have largely abandoned the jobs increasingly held by immigrants, the first diversity argument is likely insignificant in explaining aggregate labour market outcomes, while the second one has much more relevance because occupations predominantly held by natives do offer better employment prospects. Hence, as a structural feature, a large share of foreign workers in the job queue is in general expected to decrease the chances of finding employment. In this situation immigrants will benefit from looking for jobs in occupation with relatively few immigrant workers and it could therefore be a good strategy to attain credentials that gain from occupational closure in such occupations in order to avoid fierce competition. After all, the prevalence of ethnic queuing must not be overemphasized in the case of an urban labour market in which only one out of 44 occupational groups have less than 20% of immigrant workers (see Table in S2 Table).

We therefore expect that ethnic occupational segregation benefit those who are looking for jobs in which the share of foreign workers is low because job competition is relatively low in these segments (Hypothesis 2).

It is well-established that labour markets are segregated across gender [16,17,33]. In occupational labour markets segregation is preceded by a gendered system of vocational education and training which influence job placement [34]. This gendered structure may have similar effects on employment probabilities like ethnic segregation. In fact, as women are overrepresented in less desirable occupations that are not only systematically paying less than male-dominated occupations [35,36], the two forms of segregation may reinforce each other. If both groups are allocated primarily to the bottom end of the occupational hierarchy, labour supply for these jobs is artificially increased. Beyond of theirs being female or foreign, those who are looking for jobs in such occupations are exposed to higher levels of competition compared to other occupations. Female immigrants will have dismal employment prospects *because* they tend to look for such jobs. Consequently, their chances to find employment will be higher if they are trained in occupations dominated by males and/or natives. The underlying mechanism of the ethnic and gendered occupational structure is similarly related to occupational closure as job searchers—and probably female immigrant workers in particular—gain from escaping competition by attempting to avoid immigrant occupations. This is because female and male immigrants may be adversely affected by the gender divide between blue-collar and white-collar jobs [17]. Immigrant males may enjoy relatively favourable employment prospects in men-dominated blue-collar jobs such as in the skilled crafts and trades and especially in the building trades in which they increasingly supplant native workers and can expect to be paid reasonably well [33,35]. In contrast, such opportunities are not available to immigrant women apart from the low-prestigious jobs as cleaners and helpers, because labour supply for the white-collar and service sector jobs dominated by women is increasing due to ever increasing employment participation of native and immigrant women alike.

Occupational gender segregation is thus expected to benefit those who are looking for jobs which are dominated by males (hypothesis 3).

## Recent developments on the Viennese labour market

The number of working age residents (aged 15 to 64) in Vienna increased from 1.09 million in 2002 to 1.30 million in the beginning of 2018. This 19% growth in the potential labour force is entirely due to immigration. While the number of Austrian nationals in this age group has slightly decreased, the foreign population has more than doubled from 0.20 to 0.44 million [37].

Demand has not kept up with this increase in supply. Until 2016 the economic situation has been characterized by a renewed economic downturn and stagnating domestic demand

[38]. Afterwards, the Austrian economy had been gaining strength [39], until, of course, the COVID-19 pandemic changed everything for the worse. Here, we focus entirely on the pre-COVID-19 labour market, in which the increased unemployment already has widened the gap between Austrian nationals and non-nationals [40,41]. The number of unemployed nationals rose by 32% between 2002 and 2015 (from 59 to 78 thousands), while unemployment among foreigners nearly tripled from 16 to 47 thousands [42]. In 2014, 14.8% of 18-28-year olds were unemployed in Vienna and young immigrants faced a higher unemployment risk compared to natives (17.3% vs. 13.6%). This gap increased in the following years and in 2016, 20.6% of young immigrants were unemployed compared to 13.1% of natives (see Table in S4 Table). At the same time, the labour force has become more diverse, as labour immigrants from the Balkans and Turkey have been increasingly accompanied by immigrants from Eastern European countries (especially EU members since 2004) and from the Greater Middle East area (predominantly Syria, Afghanistan, Iraq and Iran). This trend of increasingly heterogeneous immigration is expected to continue [43].

## Data and methods

Ethnic employment gaps are analysed using the Labour Market Database (LMDB), which integrates process-based register data from several sources. The Social Security System records all individual spells which are covered by mandatory social insurance, among them employment, sick-days, days on parental leave, retirement and a variety of other spells. Unemployment is registered by the Public Employment Service (PES), which in addition records socio-demographic characteristics relevant to labour market placement. Besides age and sex, this includes the level and type of education attained, occupation of the aspired job, and marital status. We have access to daily records since 1997.

The LMDB thus provides timely and precise information on individual employment, unemployment and inactivity, together with background information that includes nationality of origin, which is the former nationality in case of naturalization. Using this information, we analyse employment gaps among the following origin countries: native-born Austrians, Former Yugoslav Republic (FYR, Serbia including Montenegro and Kosovo, Bosnia and Herzegovina, Croatia, North Macedonia), Turkey, Eastern European Countries (EEC, Poland, Rumania, Hungary, Slovak Republic, Bulgaria, Czech Republic), and countries of the Greater Middle East (GME, Afghanistan, Egypt, Iran, Iraq, Tunisia, Syria).

From Viennese residents aged 18-28 years, who registered as unemployed between April and September 2014, we select those who had records since 2013 (to account for previous employment experience), until 2.5 years following the start of unemployment. We exclude unemployed with a written agreement of re-employment (e.g. due to season breaks), because they typically do not look for a job. In total we have 17,463 job-seekers, about half of whom are native-born Austrians (Table in S1 Table).

### Outcome variable

The outcome measures the days it takes to find a new job of more than 90 days (i.e., relatively stable employment). Employment means dependent employment including apprenticeship training and part-time work, but excluding marginal employment (mini-jobs).

### Individual level variables

We use the highest level of education completed that is relevant to the aspired job. Seven categories are distinguished: (1) no lower secondary, (2) lower secondary, (3) apprenticeship, (4) vocational school, (5) general academic school, (6) vocational college, and (7) tertiary degree.

Education is assessed and recorded by the PES and in the case of foreign credentials it generally reflects the credential which is recognized and as such transferable to the Austrian labour market.

Prior work experience in Austria, which is crucial as on-the-job training and experience should be advantageous, is measured as the days employed in the calendar year 2013. Sample selection requires that all individuals are covered for the whole year. Four categories are used: (1) not worked at all, (2) one to 90 days, (3) 91 to 180 days and (4) more than 180 days.

Time since immigration is not measured directly. We proxy it by the first record in the register (e.g. as a dependent child) and use the following categories in years: 2-5, 6-9, 10-13, and 14 and longer. A categorical measure is used because individuals born or immigrated before 1997 are left-censored.

## Structural occupational variables

Using Labour Force Survey data (LFS), the occupational segregation measures are computed across 44 occupational categories which are a combination of ISCO minor groups (3-digits) and sub-major groups (2-digits). Form the LFS we use information on education and current or previous occupation of Viennese residents aged 18–39; see Table in S2 Table. The measures are then matched to the occupations of the aspired job as recorded by the PES. We use the aspired job information because the data does not inform us about the job actually taken. It is thus possible that the job taken actually differs from the aspired job which would affect interpretation as the occupational characteristics used in the analysis may not relate to the job found. However, the aspired job is highly relevant for the job search process guided by the PES as it is closely related to the education and the prior experience as cleared with the PES case worker. It is in fact an institutional feature of the job search because the PES provides information on vacancies and assists in applying for jobs in the aspired occupation only. Because the last job before the unemployment spell is also recorded, retrospective analysis in the following unemployment spell shows that only in exceptional cases the last job fell into another occupational group than the job aspired before this last job spell. The aspired job is thus in most cases significant for the job that was found and any deviations from that can be considered negligible.

Our closure measure reflects the extent to which occupations disperse over different credentials. We draw on the concept of mutual information [24,44], which in our case is the information obtained about the credential from knowing the occupation of a job seeker. Occupational closure can be expressed as a measure of local linkage. In analogy to Forster and Bol [25] the local linkage of one occupational group can be measured by the equation $M_j = \sum_g p_{g|j} \ln \left( \frac{p_{g|j}}{p_g} \right)$, where $p_{g|j}$ is the conditional probability of observing credential g given occupation j, and $p_g$ is the share among all workers who have attained credential g (unconditional probability). Local linkage, i.e. occupational closure, thus will be high if an occupation recruits from relatively few different credentials, and low if the occupation recruits from many different education groups. In the LFS, 34 different credential groups are considered, using a combination of level and field of the highest education attained. Another approach to measure occupational closure is the Gini concentration index. This measure has the advantage of ranging from nil (if an occupation recruits equally from all education groups) to unity (if all workers in an occupation have the same education), but it is not related to the unconditional probability. We use the Gini to check for the robustness of the findings obtained from the local linkage measure.

The other two structural variables of ethnic and gender segregation are measured as the share of non-natives, and the share of females respectively, among all workers in a given occupation. As can be seen in Table in S2 Table, the occupational structure is indeed segregated

across ethnic groups and gender, with the occupation's share of foreign workers ranging from 16.0% to 92.6% and their share of female workers from 2.9% to 92.1%.

## Control variables

Young and inexperienced workers will have dismal prospects in finding jobs, especially in times of economic slowdown when applicants queue up. Skills shortages, on the other hand, increase employment chances. We calculated the *unemployment-vacancy-ratio* for each occupation as an indicator of the structural opportunity to find employment.

Individual level controls are sex, age group in years (18-20, 21-24, 25-28) and marital status which consists of four categories: (1) single, (2) married/cohabiting, (3) divorced/separated and (4) other/missing.

## Modelling strategy

We use a simple extension of the Cox proportional hazards model. Cox regression [45] enables modelling the impact of independent variables on censored time series data, that is, where the event of interest has not been observed for all individuals. It is assumed that the estimated effect of each covariate on the event occurrence is constant over the follow-up period as compared to the baseline hazard, which is estimated from the data. To check the validity of this proportional hazards assumption the scaled Schoenfeld residuals of each variable are correlated with time. Any violation is corrected by specifying interactions with the start time which is independent of the event. Therefore, all spells are split by 91 day intervals. We use the software R [46] and its package "survival" [47]. As we observe multiple (up to four) job entries for some individuals, we use conditional models that assign the events to separate strata to allow for variation in the underlying intensity function. According to Therneau and Grambsch [48], conditional models are more efficient than modelling the time to first event only, as in standard Cox models. We limit the data to three recurrent events because less than 0.5% had more events. Time to first event analysis is performed to check for robustness.

Several models are specified. In an initial model we include gender and nationality of origin including interaction between the two to estimate unconditional gaps across countries. Then four models are estimated: Model 1 models the individual variables only, Model 2 the structural variables only, Model 3 combines the variables of Model 1 and Model 2, and Model 4 adds country fixed effects. All models include gender.

As we observe the full population of young unemployed adults, statistical inference under frequentist probability assumptions is not straightforward. Model results are descriptions of the reality established by official administration. It can make sense, however, to have an approximation about the robustness of the results to potential changes over time as a function of group sizes, because a particular pattern found in that period of time may be a historical singularity. If groups are small, joint distributional effects may change the reported relationships in the short run. We therefore include conventional standard errors in the regression tables, but the interpretation of the results focuses on the incidence (i.e. hazard) ratio as effect size [cf. 49].

## Results

We begin with a description of cumulative employment transition rates across nationalities and gender (see Fig 1). Gaps between countries tend to be larger among females (left column) than among males (right column). The upper row shows two nationalities (native Austrians and Turks) and three country groups (FYR, EEC and GME), whose nationalities are broken down in rows 2–4.

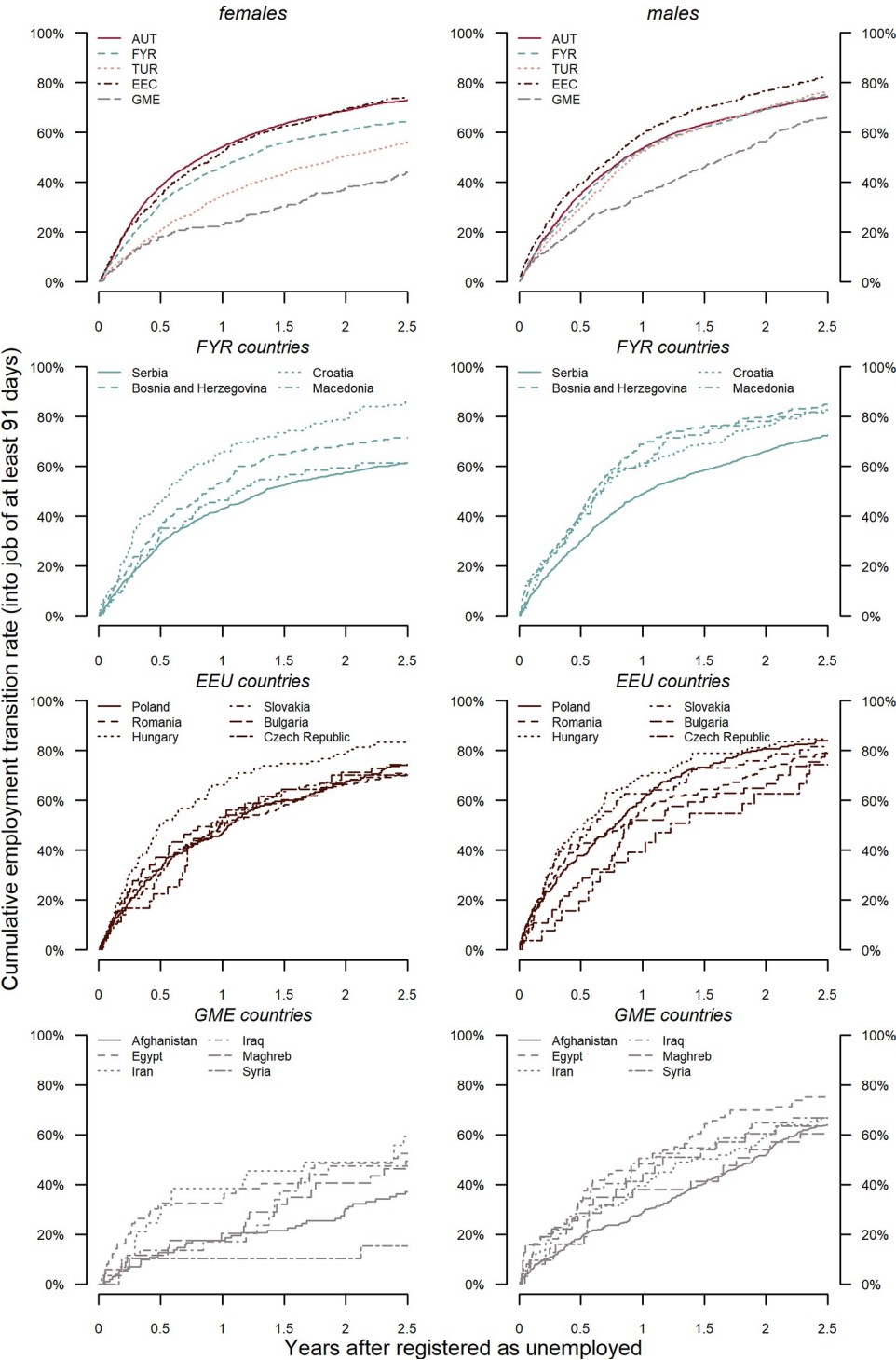

**Fig 1. Cumulative transition rates into first employment spell of young adults within 2.5 years after registered as unemployed.** Source LMDB.

Among females, the upper row reveals a hierarchy with native Austrians and Eastern Europeans at the top, followed by Former Yugoslavs, Turks and immigrants from Middle East at the bottom. Within the country groups, there is considerable variation among former

Yugoslav and in particular Middle Eastern countries, while on average there is less variation among Eastern European countries. After 2.5 years more than 80% of Croat women have found employment, which is a higher share than among native Austrians. This share cumulates to 70% of Bosnian, and 60% of Serb and North Macedonian women. Czech woman, who are slightly ahead of women from other EEC and, like Croats, outperform native Austrian women. The gaps are most pronounced among GME women. Between 50% and 60% of females from Egypt, Iran, Iraq and Tunisia have found a stable job within 2.5 years. Their trajectories, however, differ with swifter transitions observed among Egyptians and Iranians. The cumulative employment rate among Afghan women does not reach 40% and is below 20% among Syrians.

The male patterns are quite different. There is no gap between native Austrians, Former Yugoslavs and Turks in the time it takes to find employment whereas Eastern Europeans are slightly ahead, and young adults of GME origin lag behind, but gaps narrow over time. This means that while the gender gap is small within most origin countries, substantial gender disparities are observed among young job searchers from Turkey and some, but not all, GME countries.

As expected, we observe large employment gaps across ethnic groups. Nevertheless, native female job searchers on average do not outperform females from Eastern European Countries, Croatia and Bosnia, and, with the exception of some countries from the Middle East, foreign males are on par with native Austrian males.

The observed individual and structural variables explain much of the observed gaps across countries (see Table in S3 Table). Gaps among males from Iran, Iraq and Syria are fully explained, and the initially small gaps among Turks and Egyptians are further reduced. Persistent gaps are observed among Afghans and Tunisians, although in the case of the Afghans the observables explain about half of the unconditional gap. When compared to similar natives, Eastern European males are even more likely to take up employment within the observation period.

Among females, the observed characteristics explain much less of the initially larger unconditional gaps compared to males. In particular, young Syrian women are subject to huge gaps, even if they are similar to natives. Women from Iraq and Afghanistan, and to a lesser extent from North Macedonia, Turkey, Iran, Bosnia and Herzegovina, and Egypt are also less likely to find a job than native women with similar characteristics. Remaining differences across origin countries may be related to the gender role attitudes prevalent in these countries and views about the acceptability of woman's employment. If employment is in principle acceptable but restricted to certain occupational sectors and/or ethnic employers, occupational opportunities are indeed limited. Females from Tunisia are exceptional as they are on par with similar natives. On the contrary, women from Eastern EU member states do much better, and in some cases, like Hungary and Bulgaria, they do better than natives. This corresponds to lower employment probabilities among males from these countries, which points to a gender trade-off. A reversed trade-off pattern applies to Polish and Romanian job seekers, in which males tend to have higher, and females lower, employment probabilities as same sex natives.

## The impact of human capital characteristics

Model 1 includes individual human capital variables (education attainment and prior work experience) along with age, marital status and years in the country as controls. As expected, education and prior employment strongly influence the likelihood of employment (Table 1). Over the observation period young unemployed who did not finish compulsory schooling are 0.65 times as likely to find employment compared to those who have compulsory education.

**Table 1. Stratified cox proportional hazards, time to recurrent employment events.**

| | M1: individual | | M2: structural | | M3: M1 + M2 | | M4: M3 + origin countries | |
|---|---|---|---|---|---|---|---|---|
| | b (SE) | hazard | b (SE) | hazard | b (SE) | hazard | b (SE) | hazard |
| *Gender (ref: male)* | | | | | | | | |
| Female | -0.09 (0.03) | 0.92 | -0.08 (0.03) | 0.92 | -0.11 (0.03) | 0.90 | -0.05 (0.03) | 0.95 |
| *Int*: female x time (/100 days) | -0.04 (0.01) | 0.96 | -0.03 (0.01) | 0.97 | -0.03 (0.01) | 0.97 | -0.03 (0.01) | 0.97 |
| *Education (ref: lower secondary)* | | | | | | | | |
| No lower secondary certificate | -0.43 (0.06) | 0.65 | | | -0.39 (0.06) | 0.68 | -0.32 (0.06) | 0.73 |
| Apprenticeship | 0.38 (0.04) | 1.46 | | | 0.34 (0.04) | 1.41 | 0.33 (0.04) | 1.39 |
| VET school | 0.61 (0.05) | 1.84 | | | 0.60 (0.05) | 1.82 | 0.59 (0.05) | 1.80 |
| Academic School (Gymnasium) | 0.48 (0.05) | 1.61 | | | 0.45 (0.05) | 1.57 | 0.43 (0.05) | 1.53 |
| VET college | 0.75 (0.04) | 2.11 | | | 0.71 (0.04) | 2.03 | 0.68 (0.04) | 1.98 |
| University | 0.95 (0.03) | 2.58 | | | 0.83 (0.04) | 2.29 | 0.81 (0.04) | 2.24 |
| *Work experience in 2013 (ref: none)* | | | | | | | | |
| Up to 90 days | 0.39 (0.03) | 1.47 | | | 0.38 (0.03) | 1.46 | 0.36 (0.03) | 1.44 |
| 91–180 days | 0.57 (0.03) | 1.76 | | | 0.55 (0.03) | 1.73 | 0.53 (0.03) | 1.70 |
| More than 180 days | 0.75 (0.02) | 2.12 | | | 0.73 (0.02) | 2.07 | 0.71 (0.02) | 2.03 |
| *Structural variables* | | | | | | | | |
| Occupational closure M (z) | | | 0.27 (0.02) | 1.30 | 0.10 (0.02) | 1.10 | 0.10 (0.02) | 1.11 |
| *Int*: M x female | | | 0.04 (0.03) | 1.04 | 0.08 (0.04) | 1.09 | 0.07 (0.04) | 1.07 |
| *Int*: M x female x time (/100 days) | | | -0.03 (0.01) | 0.97 | -0.03 (0.01) | 0.97 | -0.03 (0.01) | 0.97 |
| Ethnic segregation (% foreign empl.) | | | -0.16 (0.03) | 0.85 | 0.02 (0.03) | 1.02 | 0.01 (0.03) | 1.01 |
| *Int*: Ethnic segregation x female | | | -0.42 (0.04) | 0.66 | -0.25 (0.04) | 0.78 | -0.23 (0.04) | 0.80 |
| Gender segregation (% female empl.) | | | -0.01 (0.02) | 0.99 | 0.00 (0.02) | 1.00 | 0.01 (0.02) | 1.01 |
| *Int*: Gender segregation x female | | | -0.26 (0.04) | 0.77 | -0.14 (0.04) | 0.87 | -0.13 (0.05) | 0.88 |
| *Individual controls* | yes | | no | | yes | | yes | |
| *Origin countries* | no | | no | | no | | yes | |
| Likelihood ratio test | 3,246 (26df) | | 1007 (10df) | | 3,374 (34df) | | 3,577 (70df) | |

Source: LMDB. *Int*: Interaction. Individual controls: Age group, years in country, marital status. Country controls are country dummies including gender interactions. M2-M4 in addition include unemployment-vacancy ratio as structural level control.

n = 112,701, number of observed events = 14,860.

In contrast, those who hold a university degree are 2.58 times more likely to find a relatively stable job at any time within 2.5 years following unemployment entry.

Independent of and beyond this large education effect, work experience is also strongly related to employment prospects. All individual variables being equal, those who worked for more than 180 days in the year prior to unemployment entry are more than twice as likely to exit unemployment as those who were not employed in 2013 at all. This large influence of the human capital variables is robust, as their estimated effects do not change much with the inclusion of the structural occupational variables in the full model (M4). A great deal of the unconditional country gaps is explained by these individual characteristics, which differ substantially across origin countries. The most frequent education among all young unemployed is a lower secondary school leaving certificate, typically awarded upon the successful completion of compulsory schooling; however, adding those who did not even reach this minimum qualification accounts for 89% of Afghan and 72% of Syrian job searchers and between 60% and 70% of immigrants from Iraq, North Macedonia, Turkey, Serbia, Romania and Egypt (see Table in S1 Table). This share is the lowest among native Austrians (34%) and Hungarians (36%). Previous employment experience on average is largest among South-eastern Europeans (238 days in

2013 among Hungarians, around 200 days among Czechs, Slovaks and Croats), followed by Poles (189 days), Bosnians (179 days) and native Austrians (175 days). Job searchers from Middle Eastern countries have acquired far less experience on the Austrian labour market, especially those from Syria (40 days), Afghanistan (68 days), while those from Iran, Iraq and Tunisia on average worked around 100 days in 2013.

### The impact of the occupational structure

Hypothesis H1 is supported by the analysis. In line with the competitive advantage argument, occupational closure is positively related to finding employment. Young unemployed who are looking for jobs in occupations that recruit from few education backgrounds are more successful than those who are looking for jobs open to more diverse education backgrounds. According to Model M2, which includes the structural variables only, an increase in the occupational closure measure of one standard deviation amounts to an increase in the likelihood of 1.24 times. Closure tends to benefit women more, as indicated by positive interaction term between closure and females. However, this holds true only in the beginning of the unemployment spell, because the negative time interaction on the female closure interaction suggests that this positive effect tends to diminish over time. Including the individual variables (Model M3) reduces the estimate of occupational closure because of its relationship with education. However, it remains substantial, in particular among females in the crucial first months of the unemployment spell, indicating that the occupational structure is credentialised beyond the education held by individual job seekers. Adding the origin country dummies (Model M4) does not affect the structural effects.

Fig 2 tries to shed light on the complex interactions of the structural variables with gender and, in case of non-proportionality, time. In it, the employment incidence rates of women and men, at the top and bottom 25% of each structural measure respectively, are contrasted to the average person in our data, pooling women and men. The predictions are based on Model M3 by using the median value for all continuous covariates and the reference category for all

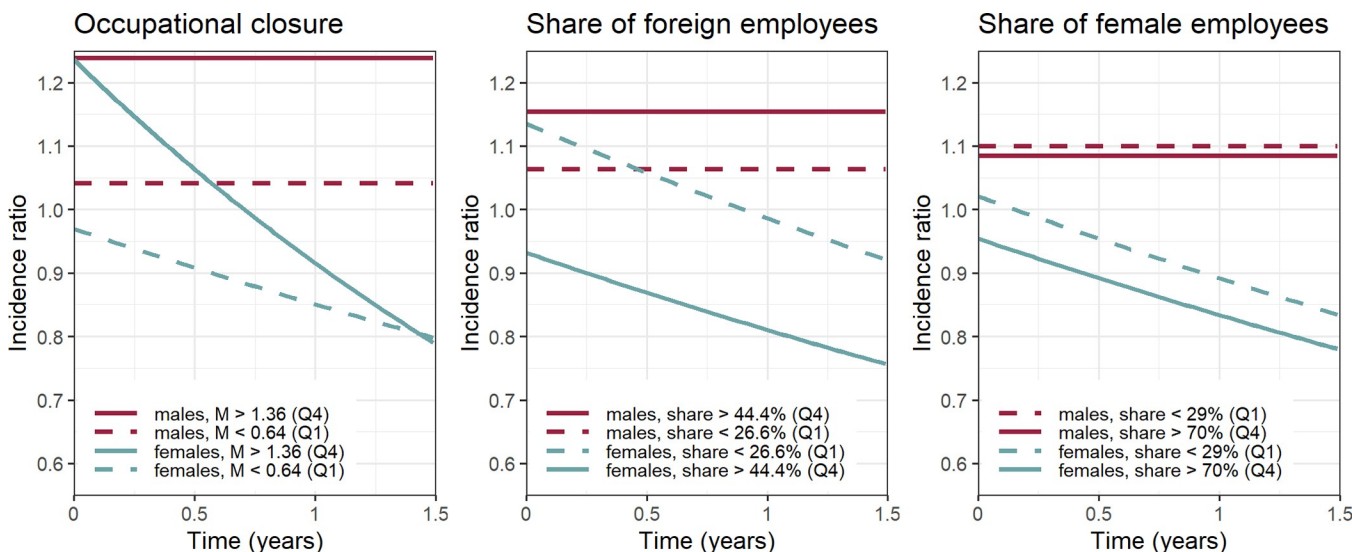

**Fig 2. Gendered occupational effects on the employment incidence.** Source: LMDB. Predicted from stratified cox proportional hazards model M3 (see Table 1), time to first employment spell of at least 91 days. Covariates were held constant at the following values: 21-24 years of age, in the country for 14 years or longer, marital status single, lower secondary education credential, more than 180 days of work experience in 2013, and median values for other structural variables.

categorical covariates. The gender differences in the incidence rates reflect the fact that the average man finds a new job faster than the average woman. The left panel confirms that a high occupational closure is beneficial to finding employment. Among males this effect is estimated to be constant over the whole follow-up period. Among females the closure advantage appears to be substantial in the first month of the unemployment spell while it decreases over time. Long-term unemployed female job seekers benefit less and after about one year in unemployment our closure measure seems to make no difference to job search success any more. However, this finding must not lead us to reject hypothesis H1 as it applies for both women and men in the crucial first months of unemployment, in which most of the transitions occur (cf. Fig 1).

Regarding ethnic occupational segregation, our results suggest that it has gendered effects. On average, males do not benefit from a higher share of native workers in their occupation. In fact, as can be seen in the middle panel of Fig 2, they tend to have better opportunities of finding a job in an occupation with a higher share of immigrant workers. In line with Hypothesis H2 young female job searchers are more successful when looking for a job in an occupation with a lower share of immigrant workers. Thus, H2 can be accepted in the case of females, whereas in the case of males the opportunities available within ethnic enterprises seem to level off the otherwise negative effect of being limited to jobs occupied disproportionally by immigrant workers. This gendered effect appears to be robust and independent of ethnic origin, as the ethnic segregation gap among females is substantial and relatively stable over time. When we exclude native Austrians from the analysis, coefficients remain stable. Including an interaction term of "foreigner" and "% foreign employees" yields a positive but statistically non-significant coefficient, suggesting that compared to native workers who look for jobs in occupations with higher shares of foreign workers, immigrant workers tend to find jobs faster in such occupations. The basic pattern holds, however: female workers, both native and foreign, greatly benefit from looking for jobs in occupations with lower shares of foreign employees. Among male workers it seems that in particular immigrants' probability to gain employment increases with the share of foreign employment while the difference among natives is small. The observation that male workers seem to benefit form ethnic segregation (contrary to what is expected in hypothesis 2) is mainly driven by increased employment prospects of immigrant workers. These additional model results are available upon request. A graphical representation of this statistically non-significant relationship can be found in the Figure in S1 Fig.

Indeed, the ethnic composition of workers varies considerably across occupations (Table in S2 Table). Immigrants are overrepresented in low paying jobs like cleaners and helpers, drivers and mobile plant operators, garment and construction workers, and underrepresented in administration, technicians, health and social workers and teachers. The finding that female immigrants benefit from looking for a job in occupations with less immigrant workers relates to gender segregation as some of the largest "female occupations" are disproportionally populated by natives, such as office clerks and secretaries, health associate professionals and hairdressers. Overall, however, there is no correlation between the occupations' shares of female and native workers.

The association between occupational gender segregation and job search outcomes mirrors that of ethnic segregation, although to a lesser extent. Occupational feminization tends to lower the employment probabilities of women, but not of men. The estimated effect is relatively small, however, and confounded by compositional effects as it is reduced greatly after accounting for individual variables. Since the proportionality assumption holds in both ethnic and gender occupational structures no time interaction with either variable has been included.

Like in the middle panel, stable gaps are thus pictured in the right panel of Fig 2, which are small among females and negligible among males.

The association between ethnic segregation and the observed gendered occupational structure seems to keep separating male and female segments in the job market. Occupations with high levels of non-native employment seem to provide job opportunities primarily to men, while woman face barriers in accessing such jobs. Recruitment practices that protect males from female competition of ethnic enterprises may contribute to the explanation of this gendered ethnic employment gap as women are less likely to enter closed "male professions". Our findings suggest that at least in the first half year of unemployment occupational closure is not gendered as it affects both men and women positively. The strategy to train for a closed occupation may thus be rational for women to escape the traps of the segregated job market. However, this strategy will not work out equally well for all women because it may not help long-term unemployed females to find their way back into work.

Additional analysis is performed to check whether our findings are sensitive to the exclusion of potential outliers and subgroups. Our results are robust to the exclusion of those occupations with the highest shares of immigrants (excluding the 9 top-quintile immigrant occupations, results see R1 in Table in S5 Table). Likewise, they are stable if the models are run exclusively on females and males, respectively. The separate gender models confirm that young women tend to benefit more from occupational closure than men and that ethnic and gender segregation primarily affects women.

## Discussion

In this study, we observe large gaps in the time to find employment between native Austrian job seekers and immigrants, and among immigrants from different origin countries. These gaps can be explained by individual and structural heterogeneity to a considerable extent, in some cases completely. However, job seekers from certain origin countries are subject to persistent gaps. With the exception of Tunisia, women from Greater Middle Eastern countries are less likely than women from other countries to find a job. We speculate that the remaining nation-of-origin effects may be related to traditional gender role attitudes prevalent among immigrants from these countries, and the views held about the acceptability of woman's employment which likely translate into limited occupational opportunities. Since the situation of males is different in most of these countries, it is not plausible that the main reason for the female gap is that employers discriminate against certain origin countries per se. Among males, only job seekers from Afghanistan and Tunisia are still less likely to find a job than natives with similar observed characteristics, but not those from other origin countries. In fact, Bosnian and North Macedonian men, along with Romanians and Croatians, tend to do better than native Austrians, who are on par with the unemployed from the other countries. Interestingly, there is a gap among males of Tunisian origin, but not among females.

The key contribution of this study is the finding that occupational labour market structures influence job-search outcomes beyond individual sociodemographic characteristics. Notably, the occupational structure appears to have gendered effects. In the aggregate men tend to benefit from ethnic occupational segregation, i.e. if they are looking for jobs in occupations with higher shares of immigrant workers while this is to the disadvantage of women, who, conversely, are more successful when immigrant shares are lower. The effect of occupational gender segregation is less clear but seems to operate to the disadvantage of women as well. Employment prospects tend to decrease with the share of females in occupational groups.

Occupational closure is positively related to job search success. Especially in the beginning of the unemployment spell, it seems to be highly beneficial for both men and women when

looking for jobs in occupations which recruits from relatively few educational backgrounds, given that they are trained and experienced in that occupation. Among females, the competitive advantage of closure may thus mitigate the negative consequences of ethnic and gender segregation. Therefore, it could be a promising strategy for young immigrant women to choose highly credentialised occupations in which the shares of immigrant workers are relatively low. However, to escape the traps of the segregated job market, this strategy will not work out equally well for all because our results suggest that the mitigating closure effect diminishes over time. In general, long-term unemployed women have an increasingly hard time to find their way back into work. Although occupational closure does not facilitate job search of long-term unemployed females, it can still make a difference to train in "native" occupations. Indeed, this may require a high degree of cultural integration and the will to adhere to the values and customs of domestic labour market segments. Females from backgrounds that hold traditional views on the duties men and women have may face barriers accruing from inconsistencies between these values and beliefs and labour market demands that require ever increasing skill and training investments. When considering future returns to on-the-job training investments, employers may prefer job applicants who do not signal traditional gender values associated with the male breadwinner model. In particular, young females before or during maternity may be subject to such a detrimental employer evaluation. Indeed, the individual job seeker is still treated unfairly if she is judged by (ascribed) group characteristics that may very well differ from her own. Anticipating such detrimental employer evaluation, minority group members may withdraw from the labour market which results in low skill levels due to deskilling and/or underinvestment in education and training. To overcome systematic gender differences in labour market outcomes associated with educational and occupational preferences necessitates changes in the attitudes and behaviour of both employers and workers.

As always, certain limitations apply to the interpretation of the results presented here. We have analysed the time it takes to find a job. Avenue for future research involve the investigation of a broader range of employment outcomes including prestige and income. Moreover, we cannot rule out that unobserved characteristics drive part of the remaining gaps. Systematic group differences in human capital endowments may exist beyond education and experience, because not all variables related to productive capacity are observed. Omitted variables to be considered in the interpretation of remaining ethnic gaps also include socio-cultural factors that may be important to consider application and hiring decisions.

## Supporting information

**S1 Fig. Ethnic occupational effects of the share of foreign employees among occupations on the employment incidence of native and immigrant workers.**
(DOCX)

**S1 Table. Descriptive statistics of individual variables.**
(DOCX)

**S2 Table. Descriptive statistics of structural occupational variables.**
(DOCX)

**S3 Table. Stratified cox proportional hazards, time to recurrent employment events.**
(DOCX)

**S4 Table. Youth unemployment rates in Vienna (18-28-year olds).**
(DOCX)

**S5 Table. Sensitivity analysis: Stratified cox proportional hazards, time to recurrent employment events.**
(DOCX)

## Acknowledgments

We thank Flavia Fossati for her extensive feedback on a prior version of this article and for very helpful suggestions.

## Author Contributions

**Conceptualization:** Stefan Vogtenhuber, Nadia Steiber, Monika Mühlböck.

**Data curation:** Stefan Vogtenhuber.

**Formal analysis:** Stefan Vogtenhuber.

**Funding acquisition:** Stefan Vogtenhuber, Nadia Steiber, Bernhard Kittel.

**Methodology:** Stefan Vogtenhuber, Nadia Steiber, Monika Mühlböck.

**Project administration:** Bernhard Kittel.

**Supervision:** Bernhard Kittel.

**Visualization:** Stefan Vogtenhuber.

**Writing – original draft:** Stefan Vogtenhuber.

**Writing – review & editing:** Stefan Vogtenhuber, Nadia Steiber, Monika Mühlböck.

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
