## [Decision Letter · Decision Letter 0]

9 Dec 2020

PONE-D-20-31176

The impact of occupational structures on ethnic and gendered employment gaps: An event history analysis using social security register data

PLOS ONE

Dear Dr. Vogtenhuber,

Thank you for submitting your manuscript to PLOS ONE. After careful consideration, we feel that it has merit but does not fully meet PLOS ONE’s publication criteria as it currently stands. Therefore, we invite you to submit a revised version of the manuscript that addresses the points raised during the review process.

Both reviewers find the topic of your article interesting and important, and note the value of the data you bring to bear on this issue.  At the same time, each reviewer offers a number of valuable suggestions that I believe will help to make the article stronger and more influential. 

Reviewer #1 identifies a number of areas in which further clarification is needed about what the data measure.  This reviewer also points out the need to more clearly explain the mechanisms that you believe underlie the phenomena you seek to analyze.  In particular, you place great emphasis on the nationality make-up of the work-force on job-finding prospects, but this is itself an endogenous outcome and depending on what factors you are focusing on could have contradictory effects.

Reviewer #2 suggests a more nuanced view of how gender-norms might operate, and how they intersect with nationality.  This reviewer also asks you to help to better contextualize the data.

Overall, I think these are excellent suggestions, and encourage you to give them due consideration should you choose to submit a revised version.  In addition, I would like to encourage you to think about making it clearer to readers much earlier on what specific question(s) you are going to answer.  The introduction and theoretical framework are currently written in very general terms and the reader must persevere to page 8 before finding out what empirical problem you are actually investigating.  Providing a clearer indication of what the article is about would make it much easier for the reader to understand and assimilate the significance of the introductory material.

We look forward to receiving your revised manuscript.

Kind regards,

Joshua L Rosenbloom

Academic Editor

PLOS ONE

Journal Requirements:

Reviewers' comments:

Reviewer's Responses to Questions

**Comments to the Author**

1. Is the manuscript technically sound, and do the data support the conclusions?

Reviewer #1: Partly

Reviewer #2: Partly

2. Has the statistical analysis been performed appropriately and rigorously? 

Reviewer #1: Yes

Reviewer #2: Yes

3. Have the authors made all data underlying the findings in their manuscript fully available?

Reviewer #1: No

Reviewer #2: No

4. Is the manuscript presented in an intelligible fashion and written in standard English?

Reviewer #1: Yes

Reviewer #2: Yes

5. Review Comments to the Author

Reviewer #1: This paper uses linked administrative data for the labor market of Vienna, Austria to examine time-to-employment for young, unemployed workers. The main interest is to consider employment gaps between immigrants and natives, and to consider gendered dimensions of these immigrant-native differences. Typically, researchers have examined individual-level skill and related characteristics – like education or time in the host country – as determinants of these gaps. This paper adds occupational characteristics: the extent to which an occupation is “closed” (hires only workers with a limited set of education or similar credentials), the extent to which it is segregated by ethnicity (or between natives and immigrants), and the extent to which it is gender-segregated.

The hypotheses generally focus on the notion that immigrants will do better (get jobs faster) in “closed” occupations and in occupations where immigrants are less present, and that female immigrants will do better in jobs where women are less present. The reasoning is that immigrants (and women) will face less competition from similar kinds of workers in these jobs.

The statistical results are varied: immigrants do find employment faster in “closed” jobs, but for women this effect diminishes with the unemployment spell. Male immigrants do not find employment faster in jobs with fewer immigrants present, but female immigrants do find employment more rapidly in jobs with fewer immigrants in them, though this effect also diminishes with the length of the unemployment spell.

I think the topic here is worthwhile and important, and the data seem really useful. I have some concerns, though, both about what is being measured and about the underlying model of employment disparity.

With regard to the data: The Public Employment Service data provide information on “desired job.” The occupational characteristics used in the analysis (% immigrant, % female, closure) appear to be connected to this desired job. The information determining time to employment comes from Labor Market Database data, and this just seems to indicate whether a job has been taken, not what that job is. So, do we know whether these workers are getting employment in their “desired job,” or just in any job? If it’s the latter, then the occupational characteristics in the analysis don’t necessarily tell us about the job that was found. How should this affect our interpretation of the results? To what degree does an indication of a “desired job” reflect something important about the job search itself? This seems to be fundamental to the argument being made here.

This seems particularly important for the “job closure” variable. The argument is that “closure” should accelerate the movement to employment, but of course this only makes sense if the individual has the required credential. Is there something institutional in this process that focuses individual job search in this way? Or is it just expected that people would not list a “desired job” if they lacked the credential for it? These issues need to

be clarified before we can really interpret the impact of occupational characteristics.

With regard to the underlying model: The above considerations apply to the analysis of % immigrant and % female in each job, but I have another concern here as well. The underlying model of employment disparity seems to be a “queuing” model, in which employers “rank” groups (men, women, immigrants, natives…) by their desirability or suitability for a job and then extend employment offers based on that ranking. So, jobs with lots of immigrants in them are jobs in which employers rank immigrants at the front of that queue, while jobs in which few immigrants are found would seem to be jobs in which employers rank immigrants at the back of that queue (see lines 153 to 158). But the argument here is that immigrants will have an easier time getting a job in an occupation with FEW immigrants present, as this will reduce competition from similar workers (lines 159-161). This seems to ignore the fact that, under the model motivating the analysis, immigrants are not in these jobs because employers deem them to be unsuitable (at the back of the queue). I don’t see how this could be taken as an indication of greater employment opportunities for immigrants in such jobs. The same argument would apply to female immigrants being predicted to have better opportunities in jobs with few women in them. This apparent contradiction in the underlying model of employment disparities needs to be explained and clarified, I think.

I also have a few minor questions and suggestions relating to presentation:

The focus on the term “employment gap” in the title and abstract is a little confusing, since the analysis is of time to employment and not the measurement of percent employed or unemployed (though obviously these things are related). I think the time-to-employment dimension of the analysis is actually unusual and really useful, so I think it would help to highlight that more in the title and abstract.

Figure 2: These figures present relative likelihoods of moving to employment, for men and women searching for occupations at the top and bottom of the distribution for the three main occupational characteristics. The baseline for comparison in all cases is the “average person in the data.” Is this the average person, pooling men and women (that is, is the baseline the same for men and women in the figures)? It seems like this must be, since men have incidence rates in excess of 1 for jobs at both the top quartile and bottom quartile of the occupational characteristic measures. Presumably this reflects the fact that men find jobs faster than women overall. This should be clarified in the discussion (or maybe graphs with gender-specific baselines should be presented?).

In line 106, I’m not sure whether “occupational labor markets” means something other than just “labor markets.” This phrase recurs throughout the paper.

The description of Hypothesis 2 in lines 159 to 161 is unclear:

159 …ethnic occupational segregation benefit those who are looking for

160 jobs in which the share of foreign workers is low because of increased competition in these

161 segments (Hypothesis 2).

Does this mean “increased competition” in segments where the share of foreign workers is *high*? The language is confusing.

Reviewer #2: Review of “The Impact of Occupational Structures on Ethnic and Gender Employment Gaps…”

The project seeks to examine the ability of unemployed young people, 18-28 years old, to secure jobs in different niches of the Austrian labor market, with the focal reference point being on or around 2013. In particular the author(s) examine three dimensions of occupations as potential influencers of securing a job – (1) social closure via occupational credentials, (2) the representation of immigrants in the occupational group, and (3) gender representation in the occupational group. The results suggest that social closure is relatively beneficial for new job seekers, especially immigrants, and that the effects of immigrant occupational composition varies by gender with male immigrants better off in immigrant niches and women better off seeking jobs where immigrants were absent.

The results are very intriguing and I have several comments about the work that are designed to make the overall presentation more effective;

(1) The data are unsurpassed in their detail and quality, but we are provided very little context for interpreting the results in the Austrian labor market. How high is youth unemployment generally and during this time period? How difficult is it to land a first job? And, overall, how gender and ethnically segregated is the labor market where our new entrants are trying to find a job?

(2) Second, it would appear that, for men, immigrants have completely supplanted locals in a few occupations and especially in the building trades. If one controls for these outliers (or excludes them from the analysis) does it change the results? The results for men seem especially susceptible to this since the blue-collar jobs that are dominated by immigrants are probably well-paying (though immigrant wages may have undercut local wages at some earlier time).

(3) Third it would appear that, for women, the opposite might be true. Immigrant women are entering occupational niches with pay that is low relative to that available to domestic women. This would make the attempt to avoid the immigrant labor market sector more understandable.

(4) In fact, points (2) and (3) lead to another potentially missing piece, that of pay and earnings inequality. From the appendix table 2, it superficially appears that immigrant men’s niche in the labor market probably pays far better than that for immigrant women. How easy or hard is it to get a job in a higher paying occupation?

There seem to be two sets of gender norms operating here, at least among immigrants in the labor market. There are clearly nation-of-origin differences in views about the appropriateness of women’s employment that create the country-level fixed effects in the analysis. Then there is the Charles/Grusky horizontal blue-collar and white collar distinction that produces extreme segregation by gender (2004, Occupational Ghettos: the Global Gender Segregation of Occupations, Stanford University Press). The horizontal distinction is called “horizontal” (rather than vertical) because many of the blue-collar jobs men dominate actually pay fairly well, require certifications to do, etc. and some of the white collar jobs that women do (and some men) would be viewed as less desirable than this blue-collar employment. So men dominate blue-collar labor markets and (in this case) if domestic men leave those labor markets the most likely replacements are immigrant men.

One last question involves the relationship between some of the ethnic groups described here and an expanded EU. So any of the skilled credentials easily migrate while others do not? I’m not familiar enough with EU labor market rules to know if this is the case, but this would affect the ability of immigrants from different places to secure good jobs.

So the analysis and presentation as of now need a bit more specificity and a policy discussion at the end about what the results say about the functioning of Austria’s labor market.

6. PLOS authors have the option to publish the peer review history of their article (what does this mean?). If published, this will include your full peer review and any attached files.

Reviewer #1: No

Reviewer #2: No

---

## [Author Response · Author response to Decision Letter 0]

26 Jan 2021

Dear Prof Rosenbloom!

Please find below our response to the very valuable and helpful comments of you and the 2 reviewers.

Following your suggestion, we have tried to make it clear very early in the introduction (lines 43-53) what the article is going to do empirically. This was missing of course, we hope this facilitates reading the article. We have also tried to make the theoretical framework more specific by referring to our empirical problem. Please read more on that below in our response to the reviewers, especially to reviewer 1. 

Response to Reviewer 1

Your point about the significance of the occupational characteristics of the “desired job” is a very good one. We address this at the beginning of the data section “Structural occupational variables” (p. 13 lines 285 - 296) where we highlight its importance for the job search process. We have changed the label to “aspired job” throughout the paper, because we believe it better reflects its purpose in the job search, although “desired job” would be the correct translation of “Berufswunsch”, which is the German term used by the PES. The PES focusses most if not all of its job search assistance on this job, which is not a mere “desire” of the individual job searcher but is closely related to the education and the prior experience as cleared with the PES case worker (it is thus also a “realistic” job to search for where the prospects are best given individual characteristics and of course also demand).

We considerably revised the theoretical section on the mechanisms underlying the structural effects in order to remedy the concerns raised. Our central argument about occupational specificity is the following: like job searchers’ aspired job belongs to a particular occupational group, these occupational groups demand more or less specific credentials. Occupational specificity thus limits competition for these jobs in the sense that it at least to some extent shelters those applicants who possess such credentials from competition from other applicants who do not have a preferred credential or a less preferred one. Except for the regulated professions however, very different credentials can apply for most jobs and we see empirically that this is the case. Now if the occupational specificity is low, competition is high as there seems to be no preferred credential but applicants from diverse educational backgrounds have good chances and actually do get such jobs. These tend to be the low-paying and low-prestige jobs (except of course managerial jobs who also have diverse educational backgrounds) and if so, they tend to be populated disproportionally by immigrants and/or females. Because the %foreign and %female structural effect mainly or exclusively applies to females (males tend to benefit from ethnic segregation and there is no effect for males regarding %females) the question about the mechanism then is why immigrant females do benefit from looking for a job in occupations with less immigrants and to some extent also in occupations with less females. We agree that this contradicts the assumption that employers deem them to be unsuitable for the job. However, we now argue more clearly that this may be outweighed by the fact that immigrants tend to apply for immigrant jobs which are located at the bottom of the labour market and where they are subject to fierce competition. In line with the queuing argument, employer may privilege native workers in order to uphold a minimum of diversity. At the top of the occupational ladder this may be the other way around: on top of less competition, skilled minorities may enjoy opportunities in order to increase diversity (p. 8-9, lines 168-190).

Likewise, we have expanded and refined our argument regarding the gendered structure thanks to your suggestions (p. 9-10, lines 205-216).

We continue to use the term “employment gap” throughout the paper but as we agree that the time-to-employment dimension is really useful, we emphasized this dimension in several parts throughout the article.

You are correct regarding Figure 2, the baseline represents the average person in the pooled data. We decided to stick with this and mentioned on p.21 lines 453-456 that “The gender differences in the incidence rates reflect the fact that the average man finds a new job faster than the average woman.”

As detailed on p. line occupational labour markets are an institutional feature of some countries, especially in continental European countries, while industrial relations in others countries, e.g. in more liberal employment regimes credentials attained in secondary schooling are of less relevance with a relatively loose linkage between schooling and work. 

Thank you for noticing the error in phrasing H2, which we corrected.

Response to Reviewer 2

We provided additional context information to aid interpretation. In the supplement (S4 table) we present recent trends in youth unemployment among our age-group (18-28 year olds) and for natives and immigrants as well. We included a text that refers to this information on p.10-11 lines 229-234. We also mention how gender and ethnically segregated the labour market actually is by referring to S2 Table (p. 14, lines 314-316).

We have also included another supplemental table (S5 table) that displays the results of a sensitivity analysis following your suggestions. We ran one model that excludes occupations with very high shares of immigrant workers (more specifically we excluded those 9 occupational groups who are in the top quintile in the immigrant share, i.e. > 48%). Two additional models were run using the data of women and men, respectively, only. We discuss these results on p.23 lines 509-515. As the results of these models are very much in line with the main findings we base our article on, we are confident that they are quite robust.

The remark about how gender norms operate is similar to the issue raised by R1. In response, we have considerably refined and extended our argument following much of your very valuable suggestions p. 9-10, lines 205-216 and in the results section on p. 17, lines 394-398 and in the discussion section on p.23, lines 522-527. Thank you for suggesting the blue-collar and white-collar distinction which we also took up not by referring to Charles and Grusky (this very interesting book seems to be unavailable in AT, even Vienna University does not hold a copy, but it is on its way from Amazon and I am looking forward to reading it) but by referring to other articles that make a similar case by following Charles/Grusky themselves.

We have also included a remark about the transferability of the credentials on p.12 lines 270-271. Our data provides information on the educational credential that is deemed to be marketable by the PES case worker.

Your valuable comments and suggestions have inspired us to be more specific in our theoretical argument as well as the discussion of our results which complements our discussion about the functioning of the Viennese labour market in p.24-25 lines 533-566.

---

## [Decision Letter · Decision Letter 1]

9 Mar 2021

PONE-D-20-31176R1

The impact of occupational structures on ethnic and gendered employment gaps: An event history analysis using social security register data

PLOS ONE

Dear Dr. Vogtenhuber,

Thank you for submitting your manuscript to PLOS ONE. After careful consideration, we feel that it has merit but does not fully meet PLOS ONE’s publication criteria as it currently stands. Therefore, we invite you to submit a revised version of the manuscript that addresses the points raised during the review process.

You have addressed almost all of the concerns raised by the reviewers, and the revised version is clearer and stronger.  The reviewers have raised a few minor issues that you should consider addressing before the article is accepted for publication.  In particular, Reviewer #1 notes an apparent inconsistency in the factors determining the hazard rate and suggests an approach to resolving it.  Reviewer #2 asks for a simple graphical representation of the relative importance of the different categories of factors explaining unemployment spells. Please consider incorporating these changes, or explain why they are inappropriate.

We look forward to receiving your revised manuscript.

Kind regards,

Joshua L Rosenbloom

Academic Editor

PLOS ONE

Journal Requirements:

Reviewers' comments:

Reviewer's Responses to Questions

**Comments to the Author**

1. If the authors have adequately addressed your comments raised in a previous round of review and you feel that this manuscript is now acceptable for publication, you may indicate that here to bypass the “Comments to the Author” section, enter your conflict of interest statement in the “Confidential to Editor” section, and submit your "Accept" recommendation.

Reviewer #1: (No Response)

Reviewer #2: All comments have been addressed

2. Is the manuscript technically sound, and do the data support the conclusions?

Reviewer #1: Yes

Reviewer #2: Yes

3. Has the statistical analysis been performed appropriately and rigorously? 

Reviewer #1: Yes

Reviewer #2: Yes

4. Have the authors made all data underlying the findings in their manuscript fully available?

Reviewer #1: Yes

Reviewer #2: No

5. Is the manuscript presented in an intelligible fashion and written in standard English?

Reviewer #1: Yes

Reviewer #2: Yes

6. Review Comments to the Author

Reviewer #1: The authors have done a nice job of addressing my original comments. In particular, they have clarified why "aspired job” is a useful category for the analysis of occupational characteristics. They now more clearly discuss their outcome measure and their underlying model. I have one remaining point of confusion regarding the model, though. In the hazard rate analysis, the "ethnic segregation" variable (which is % foreign born in the "aspired job") has a common effect on native and foreign born workers. The fact that % foreign born is expected to affect native and foreign-born workers in the same way is not clear in the development of the relevant hypothesis, hypothesis 2. For instance, we are told that (lines 183 – 185):

“Indeed, in occupations with a large share of immigrant workers employers may privilege the occasional native applicant when there is a premium on a minimum of diversity.”

But that (lines 188-191)

“… immigrants will benefit from looking for jobs in occupation with relatively few immigrant workers and it could therefore be a good strategy to attain credentials that gain from occupational closure in such occupations in order to avoid fierce competition.”

This seems to suggest that a high % immigrant in a job will affect the employment prospects of immigrants and natives differently, but a common coefficient on % immigrant, applying to both natives and immigrants, is estimated (unless I’m misinterpreting).

So, I think this point – that this occupational characteristic is expected to affect immigrants and natives in the same way – could be clarified a bit in the text. Alternatively, an interaction of “immigrant” and “% immigrant” in the estimation would clarify whether this factor affects natives and immigrants differently. An analogous interaction term is provided for the analysis of the effect of % female. In that case, we find that having an “aspired job” that is heavily female reduces the odds of exiting unemployment more for women than it does for men. The authors do indicate that they ran an analysis excluding natives and got the same results, so perhaps the effect is uniform across these two groups, but again I think that expectation could be clarified a bit in the initial discussion.

Reviewer #2: This revision answers most of my queries from the first review in a satisfactory fashion. The data used are still first rate, the statistical analysis is appropriately done, and the methdology is transparent.

If I were to make one other suggestion it would be this - I would suggest (in figure form) making a direct comparison between the effects of human capital, country of origin, and occupational structure as these are laid out in the analysis. One has to read through quite a lot of the manuscript to figure out what these relative contributions are. If I'm reading the manuscript correctly, human capital effects are the biggest by far, followed by occupational structure and then country of origin. Most of the country-of-origin effects are mediated by human capital characteristics.

So how much shorter is an unemployment spell if highliy skilled workers are looking in the "right" occupational niche? How much shorter is the unemployment spell for un-credentialled workers looking in the "right" occupational niche? A Final figure would present these relative effects nicely.

7. PLOS authors have the option to publish the peer review history of their article (what does this mean?). If published, this will include your full peer review and any attached files.

Reviewer #1: No

Reviewer #2: No

---

## [Author Response · Author response to Decision Letter 1]

1 Apr 2021

Response to Reviewer 1

We acknowledge that there may be an apparent inconstancy in the development of hypothesis 2. It is true that the two sentences lead to the expectation that native and foreign workers are affected differently by ethnic occupational segregation:

“Indeed, in occupations with a large share of immigrant workers employers may privilege the occasional native applicant when there is a premium on a minimum of diversity. The same may apply to jobs at the top where most of the workers are natives and minority workers may be welcome in order to increase diversity.” 

However, while the former mechanism may occur occasionally, the latter should outweigh it by far. To clarify this we add the sentence:

“Still, as native workers have largely abandoned the jobs increasingly held by immigrants, the first diversity argument is likely insignificant in explaining aggregate labour market outcomes, while the second one has much more relevance because occupations predominantly held by natives do offer better employment prospects.” (l184-187).

So, in fact we expect the effect as stated in Hypothesis 2, which, as we see only holds true for women. We follow your suggestion of including an interaction term of “foreigner” and “% foreign employees” as an additional robustness test. It yields a positive coefficient which is statistically insignificant (so we see no need to include it in the main paper). However, we prepared a supplementary figure based on this model (S6 figure), which suggests that the observation that male workers benefit form ethnic segregation (contrary to H2) seems to be driven mostly by foreign workers who tend to benefit more than native workers. This is discussed and described in lines 483-494.

Response to Reviewer 2

Your suggestion to include a final figure could in principle aid comprehension of the relative importance of the different factors. You are right that the two human capital variables (education and work experience) are the main drivers of reemployment probabilities. However, we would not want to include such an additional figure because our main concern are the structural effects over and above the individual human capital characteristics and thus we want to limit graphical depictions on differences in the incidence ratios related to those variables. These differences, although certainly not as large as between the highest and the lowest education level, remain the same among persons with the same education or work experience. The fact that education and work-experience have larger effects is nicely illustrated by the hazard ratios in table 1 as well as in the text. The same applies to their relative contribution in terms of model fit, as can be seen in the Likelihood Ratio tests in Table 1

---

## [Editor Report · Decision Letter 2]

7 Apr 2021

The impact of occupational structures on ethnic and gendered employment gaps: An event history analysis using social security register data

PONE-D-20-31176R2

Dear Dr. Vogtenhuber,

We’re pleased to inform you that your manuscript has been judged scientifically suitable for publication and will be formally accepted for publication once it meets all outstanding technical requirements.

Kind regards,

Joshua L Rosenbloom

Academic Editor

PLOS ONE
---

## [Editor Report · Acceptance letter]

8 Apr 2021

PONE-D-20-31176R2 

The impact of occupational structures on ethnic and gendered employment gaps: An event history analysis using social security register data 

Dear Dr. Vogtenhuber:

I'm pleased to inform you that your manuscript has been deemed suitable for publication in PLOS ONE. Congratulations! Your manuscript is now with our production department. 

Kind regards, 

on behalf of

Dr. Joshua L Rosenbloom 

Academic Editor

PLOS ONE